# Role of Purinergic Signalling in Endothelial Dysfunction and Thrombo-Inflammation in Ischaemic Stroke and Cerebral Small Vessel Disease

**DOI:** 10.3390/biom11070994

**Published:** 2021-07-06

**Authors:** Natasha Ting Lee, Lin Kooi Ong, Prajwal Gyawali, Che Mohd Nasril Che Mohd Nassir, Muzaimi Mustapha, Harshal H. Nandurkar, Maithili Sashindranath

**Affiliations:** 1Australian Center for Blood Diseases, Central Clinical School, Monash University, Alfred Hospital, Melbourne, VIC 3004, Australia; natasha.lee@monash.edu (N.T.L.); harshal.nandurkar@monash.edu (H.H.N.); 2School of Pharmacy, Monash University Malaysia, Subang Jaya 47500, Selangor, Malaysia; ong.linkooi@monash.edu; 3School of Biomedical Sciences and Pharmacy, University of Newcastle, Callaghan, NSW 2308, Australia; 4Priority Research Center for Stroke and Brain Injury, University of Newcastle, Callaghan, NSW 2308, Australia; 5Hunter Medical Research Institute, New Lambton Heights, NSW 2305, Australia; 6NHMRC Center of Research Excellence Stroke Rehabilitation and Brain Recovery, 245 Burgundy Street, Heidelberg, VIC 3084, Australia; 7Faculty of Health, Engineering and Sciences, School of Health and Wellbeing, University of Southern Queensland, Toowoomba, QLD 4350, Australia; Prajwal.Gyawali@usq.edu.au; 8Department of Neurosciences, School of Medical Sciences, Universiti Sains Malaysia, Kubang Kerian 16150, Kelantan, Malaysia; nasrilche123@gmail.com (C.M.N.C.M.N.); mmuzaimi@usm.my (M.M.)

**Keywords:** CD39, endothelial dysfunction, brain, stroke, endothelial cells, purinergic signaling

## Abstract

The cerebral endothelium is an active interface between blood and the central nervous system. In addition to being a physical barrier between the blood and the brain, the endothelium also actively regulates metabolic homeostasis, vascular tone and permeability, coagulation, and movement of immune cells. Being part of the blood–brain barrier, endothelial cells of the brain have specialized morphology, physiology, and phenotypes due to their unique microenvironment. Known cardiovascular risk factors facilitate cerebral endothelial dysfunction, leading to impaired vasodilation, an aggravated inflammatory response, as well as increased oxidative stress and vascular proliferation. This culminates in the thrombo-inflammatory response, an underlying cause of ischemic stroke and cerebral small vessel disease (CSVD). These events are further exacerbated when blood flow is returned to the brain after a period of ischemia, a phenomenon termed ischemia-reperfusion injury. Purinergic signaling is an endogenous molecular pathway in which the enzymes CD39 and CD73 catabolize extracellular adenosine triphosphate (eATP) to adenosine. After ischemia and CSVD, eATP is released from dying neurons as a damage molecule, triggering thrombosis and inflammation. In contrast, adenosine is anti-thrombotic, protects against oxidative stress, and suppresses the immune response. Evidently, therapies that promote adenosine generation or boost CD39 activity at the site of endothelial injury have promising benefits in the context of atherothrombotic stroke and can be extended to current CSVD known pathomechanisms. Here, we have reviewed the rationale and benefits of CD39 and CD39 therapies to treat endothelial dysfunction in the brain.

## 1. Introduction

The endothelium is a monolayer of cells that is able to respond to physical and chemical signals by the production of a wide range of factors that regulate vascular tone, cellular adhesion, thromboresistance, smooth muscle cell proliferation, and vessel wall inflammation [1]. It also acts as a signal transduction interface to generate appropriate vasomotor signals such as ion channel activation, biochemical, and transcriptional events [2].

The endothelium shows large phenotypic heterogeneity in different vascular sections and organs including variations in morphology, behavior, and biosynthetic repertoire [3] due to differing needs specific to the local environment. Most organs such as the brain as well as the blood vessels of the circulatory system show a continuous endothelial monolayer with a well-expressed continuous basement membrane consisting of matrix components such as laminin, fibronectin, vinculin, and collagen expressed by endothelial cells during vascular growth [2].

The vasculature of the brain is made up of a large network of small and large vessels, the entirety of which is lined with endothelial cells (ECs). Brain capillary ECs lack fenestrations, are interspersed by tight junctions, and have reduced pinocytotic activity [4]. Additionally, the endothelium of the cerebral vessels serves a special function: they are a critical structural component of the neurovascular unit that constitutes the blood–brain barrier (BBB). It is the primary barrier between systemic circulation and the brain, and it regulates the transportation of solutes between the blood and brain [5]. The cerebral vascular endothelium has an overall prothrombotic-antifibrinolytic environment characterized by reduced levels of both thrombomodulin and tissue plasminogen activator (tPA) and high levels of plasminogen activator inhibitor-1 (PAI-1) [4].

This review aims to discuss endothelial dysfunction as the underlying cause for ischemic stroke and CSVD, and to expand on the cytoprotective potential of the purinergic signaling pathway in these conditions.

## 2. Endothelial Dysfunction

The endothelium has a key role in the production and release of nitric oxide (NO), which is produced from arginine by eNOS (endothelial nitric oxide synthase). NO relaxes smooth muscle cells, thereby preventing leukocyte adhesion and migration into the arterial wall. NO also curbs muscle cell proliferation, platelet adhesion and aggregation, and expression of adhesion molecules such as vascular cell adhesion molecule (VCAM) and intercellular cell adhesion molecule (ICAM). Known cardiovascular risk factors like aging, smoking, diabetes, hypertension, and high homocysteine or cholesterol levels cause structural and functional disturbances to the endothelium, leading to endothelial dysfunction. This occurs due to NO bioavailability being limited either by reduced eNOS production or by increased release of reactive oxygen species (ROS). Together with impaired NO production and altered vasodilatation, damaged endothelial cells secrete vasoconstrictors including endothelin-1, thromboxane A2, prostaglandin H2, and promote more ROS release. Endothelial dysfunction also initiates inflammation and apoptosis of endothelial cells and adjacent cells in the vessel wall [6].

### Emerging Role of Erythrocytes in Endothelial Dysfunction

Red blood cells also play a role in endothelial dysfunction, though is largely under researched. The biconcave shape of erythrocytes is of great physiological significance as it confers a larger surface area relative to volume and is important for an effective gaseous exchange. Furthermore, this allows the drastic shape change that erythrocytes need to undergo while traversing through capillaries [7]. The diameter of the capillaries that erythrocytes need to traverse are often smaller than the diameter of the erythrocyte (7 µm) itself. In such cases, erythrocytes deform to take a parachute-like shape, which facilitates their transit across capillaries, a phenomenon known as erythrocyte deformability. Reactive oxygen species (ROS) release during ischemia damages the membranes of erythrocytes, resulting in increased fragility and reduced maturation and erythrocyte distribution width, which decreases erythrocyte deformability. Rigid erythrocytes tend to aggregate [8], which increases whole blood viscosity [9] and underlies vessel wall shear stress. Shear stress is a mechanical force acting directly on ECs, and disturbances in mechanical forces on ECs affect the NO-synthesizing mechanisms in the vascular endothelium [10]. Thus, erythrocyte deformability is an important determining factor for normal microcirculation. Decreased erythrocyte deformability causes microcirculation sludging as it is difficult for a rigid erythrocyte to squeeze in through the capillaries [11,12].

Microcirculation sludging also induces endothelial dysfunction as it increases the interaction of blood cells and plasma constituents with the endothelium [13]. The interaction of rigid, aggregated, and hyper-viscous erythrocytes with the capillary endothelium results in structural and biochemical changes, altering the microvascular integrity [14,15,16], which further promotes endothelial dysfunction [17,18]. Indeed, microscopic analysis has shown focal endothelial swelling and loss of pinocytic vessels in the affected cells [19]. In addition, microcirculation sludging decreases wall shear stress in the endothelium and downregulates the NO-dependent flow mediated vasodilation [10,20,21]. An observation in a rat model of erythrocyte aggregation showed blunted flow mediated dilation and suppressed endothelial NO synthesis mechanism [10]. Decreased erythrocyte deformability may also induce endothelial dysfunction by decreasing bioactive endogenously synthesized NO [22]. A reduced concentration of NO promotes vasoconstriction, platelet aggregation, smooth muscle cell proliferation and migration, leukocyte adhesion, oxidative stress, and systemic inflammation [23,24], all leading to thrombosis.

Microcirculation sludging beyond a certain threshold triggers thrombotic events [25]. Erythrocytes can contribute to thrombin generation independent of platelets and leukocytes [26,27]. This was attributed to the presence of an exposed phosphatidyl serine (negatively charged phospholipid, normally on the cytoplasmic side of the membrane) on a very small number (0.51%) of the erythrocyte population even under physiological conditions. The thrombin generation potential of erythrocytes is linearly increased corresponding to the increase in hematocrit [28]. Microcirculation sludging increases local hematocrit, which creates a favorable environment for thrombin generation. Reduced blood flow due to an experimental increase in erythrocyte aggregation and whole blood viscosity has been shown to rapidly create thrombosis (within 30 min) at the site of endothelial injury in rabbit femoral veins [29]. Erythrocytes also adhere to the endothelium through adhesion molecules such as VCAM-1, and such adhesion is associated with a higher risk of thrombosis and vascular complications [30]. In summary, altered hemorheology (i.e., decreased erythrocyte deformability, increased erythrocyte aggregation, and increased whole blood viscosity) increases the capillaries’ flow resistance and microcirculation sludging, causing endothelial dysfunction [31,32]. Indeed, altered hemorheology is an independent risk factor of primary and secondary stroke and may contribute to stroke pathogenesis.

## 3. Thrombo-Inflammation in Atherothrombotic Stroke

The brain has high energy demand and accounts for ~20% of total oxygen consumption, which makes it highly susceptible to oxidative stress [4]. The endothelium is the regulator of hemostasis in the brain [33,34] and oxidative stress causes endothelial activation leading to a pro-thrombotic and pro-inflammatory phenotype and a loss of BBB integrity. Changes to the vascular structure and mechanics also contribute to endothelial dysfunction [34], for example, degeneration of vascular wall and loss of BBB integrity promotes movement of molecules out of circulation, and into the parenchyma [35,36].

Endothelial dysfunction affecting cerebral blood flow is a significant underlying cause of both ischemic stroke and CSVD. Ischemic stroke occurs either due to large vessel arterial thrombi or due to cerebrovascular complications affecting both large and small blood vessels of the brain. CSVD is characterized by microvascular changes, resulting in small regions of ischemia and microbleeds (microhemorrhages) [4].

### 3.1. Cerebral Ischemia Leading to Stroke

Acute ischemic stroke (AIS), accounting for approximately 85% of stroke episodes [37], results from transient or permanent reduction in cerebral blood flow (cBF) of a major brain artery [38,39]. This event is caused by an occlusion of a cerebral artery due to thrombosis or an embolism, which stems from underlying endothelial dysfunction [37,38]. It is important to understand this complex and damaging neurological condition, as stroke is the leading cause of death and acquired disability worldwide [37,40].

#### Mechanism of Endothelial Dysfunction in Ischemic Stroke

Endothelial dysfunction largely underpins the pathogenesis of ischemic stroke. In response to ischemia and its subsequent hypoxic state, there is a severe restriction of oxygen and glucose delivered to the brain [38,39]. The lack of blood flow also causes increased edema via intracellular Ca^2+^ excess [38,41]. Further, ischemia impairs endothelium-dependent vasodilation, which limits reperfusion of blood flow [34]. Impaired endothelium-dependent vasodilation of cerebral blood vessels is associated with several key risk factors for stroke such as chronic hypertension, diabetes, and hypercholesterolemia [42].

Impairment of endothelial nitric oxide synthase (eNOS) is also implicated in ischemia-induced endothelial dysfunction. Lack of eNOS activity leads to sharp increases of ROS due to impaired NO generation [43]. Increased generation of ROS by the damaged endothelium and subsequent hypoxia triggers pro-inflammatory gene activation to activate microglia and astrocytes, and causes an influx of inflammatory cells [38,44,45] recruited by cytokines, adhesion molecules, and chemokines [37]. The endothelium is further dysregulated due to the influx of inflammatory cells into the brain parenchyma. Ischemic stroke induces a strong inflammatory response that begins as soon as a few hours of initial onset, and it especially characterizes the delayed response to ischemia [37]. Ischemic inflammatory damage to the brain involves the infiltration of neutrophils that cause microvascular obstruction, and worsens ischemic injury and endothelial dysfunction [46], and the production of toxic mediators by activated inflammatory cells and injured neurons (i.e., cytokines, NO, superoxides) that can amplify endothelial damage [38,41]. Furthermore, neutrophil stalling of brain capillaries contributes to reperfusion failure [47] (Figure 1).

In cerebral ischemia, the accumulation of ROS and impairment of eNOS generation propagates a pro-inflammatory and pro-oxidant environment [43]. This potentiates thrombus formation and further increases in proinflammatory cytokines [i.e., interleukins (IL-6 and IL-8, TNF-α, and monocyte chemoattractant protein 1 (MCP-1)], and expression of vascular adhesion molecules triggering endothelial dysfunction [48].

BBB disruption and edema is also a key contributor of endothelial dysfunction [38,41], where damage to the brain causes an activation of inflammatory response initiated by neutrophil infiltration through the BBB [38,44,45]. Reperfusion is denoted by the reestablishment of blood flow to the ischemic area, and though reperfusion is necessary for tissue survival, it contributes significantly to additional tissue damage and vessel permeability, which is termed ischemia-reperfusion injury. Increased neutrophil infiltration has been shown to correspond to the biphasic permeability of the BBB [49] and initial reperfusion can also account for the loss of cerebral autoregulation and reactive hyperemia. Following the initial hyperemia, hypoperfusion of the ischemic area occurs, which can enhance neutrophil adhesion, oxidative stress, and subsequent inflammatory activity, and directly contributes to the subsequent vessel permeability [50].

### 3.2. Cerebral Small Vessel Disease (CSVD)

CSVD of the brain parenchyma represents a spectrum of clinical and neuropathological processes with various etiologies involving the cerebral microvasculature (50–400 µm in diameter) that penetrates and supplies the brain subcortical region [51,52]. This microcirculation network includes small penetrating arteries (chiefly middle cerebral artery tributaries), arterioles, capillaries, small veins, and venules. With increasing prevalence amongst older people, CSVD is a significant precursor for dementia and stroke. Manifesting as lacunar infarcts, microbleeds, enlarged perivascular spaces, and cortical atrophy, CSVD accounts for almost 45% of the total global cases of dementias, 25% of ischemic strokes, and over 70% of vascular dementia [53,54]. Given the neuroimaging identification of several manifestations of CSVD, varying degree of prevalence and health burden have been reported in both healthy and diseased populations [55,56,57].

#### Evolving Pathomechanism of CSVD

It is recognized that most of the macrostructural manifestations in CSVD as detected from neuroimaging are likely to be the end-result of the underlying mesostructural responses such as cerebral microcirculation flow obstruction (either from intrinsic or extrinsic causes). In this instance, the arteriolar occlusion or narrowing results in a symptomatic ischemic event as seen in small lacunar infarcts in the classical CSVD spectrum. Axonal injury, neuronal apoptosis, demyelination, and oligodendrocyte damage also occur during CSVD, leading to cerebral parenchyma damages, which may manifest as neurological symptoms, clinical signs, and multifaceted neuroimaging findings including silent brain infarct [58]. The pathomechanism of sporadic CSVD also encompasses systemic dysregulations, which include coagulopathy, elevated microvascular thrombosis, increased cellular activation, inflammation, and oxidative stress. These processes influence the underlying cerebral microstructural changes known to occur in CSVD (i.e., endothelial dysfunction, altered cBF, and breakdown of the BBB, Figure 1).

## 4. Involvements of Purinergic Signaling

### 4.1. Purinergic Signaling

Purinergic signaling is the collective term to describe how purine nucleotides function as extracellular signaling molecules. The adenosine-signaling pathway constitutes adenosine triphosphate (ATP), adenosine diphosphate (ADP), adenosine monophosphate (AMP), and adenosine, which modulate a range of functions such as EC growth and apoptosis, coagulation, vessel tone, and inflammation. ATP is hydrolyzed into ADP and subsequently into AMP by ectonucleotidase CD39; AMP is then converted into adenosine by CD73, which is ubiquitously expressed (Figure 2). Purinergic signaling is involved in the regulation of the immune responses [59]; ATP release in response to inflammatory mediators was found to be a basic mechanism required for neutrophil activation and immune defense [60]. The receptors P1 and P2 expressed on the surface of immune cells are activated by adenosine and ATP, respectively, and mediate the immunomodulatory effects of purines. Four G protein-coupled P1 or adenosine receptors have been identified as A_1_, A_2A_, A_2B_, and A_3_, whereas the P2 receptors have been classified into two subfamilies: the ionotropic P2X (P2 × 1–7) and metabotropic P2Y (P2Y1,2,4,6,11–14) receptors [61,62].

Cellular ATP is the main energy source of the body that drives virtually all cell functions. Intracellular ATP is involved in energy transport and pathophysiological roles, while extracellular ATP (eATP) serves as a multifunctional intercellular signaling system and is ubiquitously used for cell–cell communication [63]. Not only is eATP actively secreted through plasma membrane proteins such as pannexin and connexin, it is also passively released after stress induced cell injury [64]. Particularly, eATP levels are increased during times of tissue stress, for example, during necrosis, apoptosis, hypoxia, or inflammation [65] and acts as danger-associated molecular pattern signals (DAMP) to bind to purinergic receptors and to induce an inflammatory signaling cascade [61,66]. ATP is present in high concentrations in the brain [67] due to large energy requirements. ATP concentrations can rise due to vesicular release from neurons, exocytosis from astrocytes and microglia, or through membrane channels from glial cells and pyramidal neurons [68]. During ischemia-hypoxia, eATP and adenosine increase drastically due to glutamate accumulation from ion pump failure [37,38], exocytosis from microglia [69], and astrocytes [70], among other reasons. This excessive production of eATP and adenosine is high enough to stimulate all nearby P1 and P2 receptors [71], which are present at significant levels in neurons and most peripheral inflammatory cells [68]. This in turn induces a pro-inflammatory environment. In fact, eATP itself aggravates cerebral damage after ischemic stroke as high ATP concentrations induce cell death [72].

ADP is a platelet agonist that functions in physiological hemostasis and thrombosis [73]. It is an important mediator of platelet activation induced by other activators such as thrombin and collagen, which in turn, creates a positive feedback loop promoting ADP release from intraplatelet storage pools, thereby enhancing platelet aggregation and the proliferation of a platelet plug [73] during injury.

Adenosine, which is produced by the breakdown of ATP, is a key endogenous molecule that regulates tissue function [74] and acts as a endogenous neuroprotectant [68]. Specifically, it triggers IL-10 production [75], which suppresses cytokine secretion, antigen presentation, and CD4^+^ T cell activation [76]. Adenosine accumulates in the extracellular space in response to metabolic stress and cell damage, and hypoxia, ischemia, and inflammation all stimulate local adenosine productions [74]. The rapid release of adenosine in response to noxious stimuli has a dual role in modulating homeostasis. First, it reports tissue injury in an autocrine and paracrine manner to surrounding tissue, and it generates organ protective tissue responses, thereby restoring homeostasis [77]. Extracellular adenosine dampens hypoxia-induced inflammation [78] and pharmacologically increasing the breakdown of ATP to adenosine is effective in attenuating tissue injury and sterile inflammation during ischemia-reperfusion injury [79,80,81,82,83,84,85]. Adenosine also has an important function in suppressing the immune response, inhibits T-cell activation, and adenosine receptors (A_2A_ receptors) are found on CD4^+^ T cells [86].

### 4.2. Role of Endothelial CD39 in Preventing Thrombo-Inflammation

CD39 is an ecto-nucleoside triphosphate diphosphohydrolase (NTPDase) expressed by endothelial cells in the vasculature. Its primary action is to convert ATP/ADP into AMP, and subsequently CD73 converts AMP into adenosine (Figure 2). CD39 is liberated from the coronary vascular endothelium by ischemia-reperfusion, and levels of circulating ectonucleotidase may reflect the severity of ischemic vascular injury [87]. Attenuation or lack of CD39 activity is associated with vascular dysfunction and remodeling in pulmonary arterial hypertension [88] and failure of vascular reconstitution [89].

CD39/CD73 regulates the vascular environment by converting the pro-thrombotic and pro-inflammatory ATP and ADP rich environment into an antithrombotic anti-inflammatory adenosine rich state [65] through platelet aggregation, tissue factor expression suppression, and anti-inflammatory mediation [90,91,92].

CD39 has an integral role in normal hemostasis, and regulates prothrombotic and proinflammatory responses [93]. Expression of CD39 is a key determinant of occlusive thrombus formation and therapies that promote CD39 expression are likely to increase antithrombotic efficacy. Global overexpression of human CD39 yields profound antithrombotic effects [94,95,96], though its effect may in part be due to the reduction of ADP-induced platelet activation [97,98]. CD39 is also important in leukocyte trafficking across the endothelium in response to chemokines and regulates immune cell adhesion to the endothelial layer where cell adhesion is promoted by the ATP rich environment and inhibited by adenosine [99]. CD39^−/−^ mice display increased leukocyte adhesion as well as impaired adenosine generation, resulting in increased EC activation, greater monocyte recruitment, platelet aggregation, and increased endothelial permeability [92,100,101,102].

### 4.3. Involvement of Adenosine as Potential Innate Neuroprotection

Adenosine has an important role in the brain’s response to an ischemic event. Studies have demonstrated that the extracellular adenosine concentration levels in the brain increase within the first 20 min after ischemia and return to basal after 4 h [71]. This initial burst of adenosine may represent a critical biological protective feature of the brain following a stroke. Pignataro et al. showed an increase in infarct volume after transient middle cerebral artery occlusion [103] in transgenic mice that had low levels of adenosine due to overexpression of adenosine kinase, a main metabolic enzyme of adenosine [104].

In humans, an increase in circulating plasma adenosine levels, peaking on day 2 for transient ischemic attack and day 3 for stroke [105], further supports the role of adenosine as an endogenous innate self-repair neuroprotective mechanism. Its role in modulating systemic immune responses after stroke is also noteworthy [106]. The peripheral immune cells express adenosine receptors and respond to changes in extracellular adenosine. Adenosine modulates the activity of immune cells differentially depending on the receptors that are activated. For instance, the activation of A_2A_ and A_2B_ receptors stimulates an anti-inflammatory response by the peripheral immune cells. This property could potentially be exploited to reduce inflammatory damage in the CNS caused by infiltrating activated immune cells after stroke. In the other hand, the activation of A_1_ receptors on immune cells promotes migration and phagocytic activity [107] as well as angiogenesis through their interaction with vascular ECs [108].

Logically, pharmacological agents that modify local concentration of ATP and adenosine have been explored as a protective strategy after stroke. Therapies promoting adenosine generation are beneficial in animal models of stroke but have not been successful clinically. Adenosine is neuroprotective; administration of adenosine reduces both stroke-related brain damage and hypoxic–ischemic neuronal injury. Indeed, Kitagawa et al. showed that exogenous adenosine infusion into the ipsilateral striatum (immediately after occlusion, and continued until 3 h after reperfusion) significantly reduced infarct volume and improved neurological outcome after transient middle cerebral artery occlusion [103]. However, controlled, targeted delivery of adenosine to the central nervous system is not feasible as its half-life in human plasma is <5 s. Adenosine receptor agonists overcome this issue but can only be delivered at early timepoints after ischemia as delayed administration is associated with systemic off-target effects [68].

One therapeutic approach is to enhance hydrolysis of eATP by upregulation of nucleoside triphosphate diphosphohydrolases (NTPDases) and ecto-5′-nucleotidase, thus inducing adenosine formation. A few of the more promising therapies include Activin A, APT102, and HMG-CoA reductase inhibitors. Activin A is a pleiotropic cytokine used to upregulate the expression of CD39, and to stimulate CD39 related anti-inflammatory processes. Mukerji et al. demonstrated that intracerebroventricular administration of Activin A exerted neuroprotective effects including reduced infarct volume and improved neurological outcomes in experimental stroke. Activin A treatment attenuated neuronal apoptosis pathways as well as microglial activation [109]. A follow up study by Buchthal et al. demonstrated that intranasal administration of Activin A reduced infarct volume [110] after stroke. Another promising molecule is APT102, a recombinant form of CD39 human apyrase. Tan et al. demonstrated that the combination of r-tPA (tissue plasminogen activator) and APT102 reduces mortality rate, decreases infarct volume, and improves neurological deficit scores as well as prevents r-tPA-induced hemorrhage transformation in the experimental stroke model [111]. An advantage is that APT102 does not act directly on platelets but has an antithrombotic effect as a result of hydrolyzing eADP and inhibiting platelet aggregation with a minimally elevated risk for bleeding. Interestingly, statin (or HMG-CoA reductase inhibitors), commonly used cholesterol-lowering drugs, have been shown to increase endothelial CD39 expression [112] and reduce thrombin-induced platelet aggregation [113], suggesting antithrombotic and anti-inflammatory effects on vascular cells. Two meta-analyses have shown that statin therapy at stroke onset is associated with good functional outcome in patients with ischemic stroke [114,115]. Nevertheless, therapies that target adenosine generation should be viewed with caution due to the widespread distribution of adenosine receptors within the brain. Furthermore, the optimal window of administration for adenosine-modulating drugs is critical as the complex pathophysiological processes following stroke are influenced differentially by adenosine both spatially and temporally.

## 5. Therapeutic Potential of CD39

CD39 and CD73 are the principal enzymes involved in endogenous adenosine generation. Initial studies based on CD39 knockout mice and pharmacological inhibition of CD39 both showed that it is a crucial enzyme for protecting against increased vascular permeability and neutrophil extravasation during local hypoxia [101,116,117]. In CD39 knockout mice, cerebral [92,102], cardiac [85,118], renal [81,119], and intestinal [83] injuries were more severe than wildtype littermate controls. Pharmacological adenosine receptor agonists, or administration of soluble CD39 before the induction of ischemic insult induced a protective effect in the wild-type and the knockout mice. Pinsky et al. showed that CD39 knockout mice exhibited increased cerebral infarct volumes, and decreased post ischemic perfusion, and exogenous addition of soluble recombinant CD39 was able to reduce thrombosis, increase post-ischemic perfusion, and decrease cerebral infarct volumes in vivo even when given 3 h post-stroke. They also observed that the exogenous addition of CD39 did not cause any bleeding complications [102].

The protective role of CD39 has also been validated using transgenic mice that overexpress CD39. In a mouse cardiac transplant model of vascular rejection, it was shown that the transgenic mice were also substantially protected from thrombosis and survived longer post-transplant [120]. We showed that overexpression conferred protection in a murine model of warm renal ischemia reperfusion injury, and in a murine model of liver transplantation [121,122]. Interestingly, the beneficial effect did not seem to be due to elevated levels of CD39 in the liver, but rather by a CD39-mediated reduction in CD4^+^ cells in the donor liver [121], confirming that CD39-mediated adenosine generation is important in regulating the immune response following ischemia-reperfusion injury.

The therapeutic potential of CD39 in the ischemic brain was explored by Baek et al. [123], who developed a transgenic mouse overexpressing human CD39 either globally or in myeloid-lineage cells only. They found that the transgenic CD39 mice had decreased recruitment of macrophages and neutrophils, and overall decreased inflammation in the ischemic hemispheres. These mice also exhibited reduced neurological deficit and smaller infarct volumes, suggesting an overall therapeutic benefit of CD39 in stroke [123].

Previous work has demonstrated that in other CD39 transgenic mice, there was no overt spontaneous bleeding tendencies under normal circumstances. However, they did exhibit impaired platelet aggregation, prolonged bleeding times, and resistance to systemic thromboembolism [96,121,124].

CD39 and CD73 promote ATP hydrolysis to adenosine and shift the balance toward immunosuppressive microenvironments. Being the rate-limiting enzyme in this process, CD39 based therapeutics have the added advantage of ATP hydrolysis as well as promoting sustained adenosine generation. However, CD39 is also a potent antithrombotic and can cause bleeding if given systemically, but targeted-CD39 therapies promote localized, sustained adenosine generation at the site of cellular injury without systemic bleeding effects.

## 6. Targeted CD39 Therapeutics for Ischemic Stroke

This concept of targeted CD39 delivery is demonstrated by Hohmann et al. [125], where they aimed to target CD39 to decrease ADP-induced platelet activation. Soluble CD39 had previously been shown to reduce ADP concentration, preventing platelet activation and recruitment [126]. The group aimed for a strong antithrombotic effect via accumulation of CD39 through targeting with a single-chain antibody (scFv) against the active conformation of GPIIb/IIIa found on activated platelets. Their compound, αIIbβ3Targ-CD39, exhibited specific binding to activated platelets and successfully inhibited the thrombosis of the carotid artery at a CD39 dose, approximately 10 times lower compared to a non-targ-CD39, without prolonged tail-bleeding time, which has been the main limiting factor in CD39 therapies [125].

This concept was further explored by our group; we showed that a novel construct of recombinant soluble CD39 (rsol.CD39-PSGL) linked with a 20 amino-acid P-selectin tag was able to protect against kidney IRI at a dose that did not perturb hemostasis. The mice injected with the construct showed significantly less kidney damage compared to those injected with the soluble CD39, and that a lower dose of targeted CD39 was needed as opposed to soluble CD39 to confer protection, without prolonged tail-bleeding time [127].

The efficacy of targeted CD39 against thrombosis was tested when a novel construct of dimeric GPVI-Fc was fused to CD39 (GPVI-CD39), aimed to increase platelet inhibitory potential and to create a lesion-directed dual antiplatelet therapy with minimal bleeding risks [128]. It was found that there was significant inhibition of ADP-induced platelet aggregation as well as significantly delaying ferric chloride induced thrombosis, while not increasing bleeding potential [128].

Therefore, targeting CD39 to either activated platelets of a thrombus or to an activated endothelium is a potent antithrombotic, and can be delivered in systemic concentrations that avoid the risk of bleeding, and is thus a potential therapeutic for vascular diseases.

## 7. Conclusions

Endothelial dysfunction in the brain is an important underlying pathology of ischemic stroke and cerebral small vessel disease. It promotes oxidative stress, alters erythrocyte deformability, and promotes platelet activation and thrombosis as well as adhesion and transmigration of leukocytes. Adenosine is an endogenous molecule that can attenuate each of these pathological events. Accordingly, CD39 and CD39 therapies, that boost local adenosine levels in the brain, are strong therapeutic candidates to treat endothelial dysfunction underlying cerebrovascular diseases.

## Figures and Tables

**Figure 1 biomolecules-11-00994-f001:**
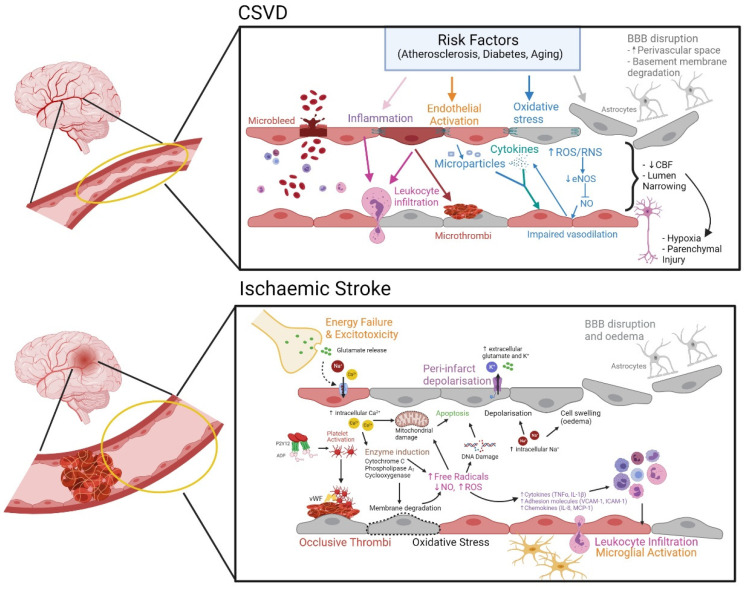
Endothelial dysfunction underlies the pathophysiology of CSVD and ischemic stroke. **CSVD**: Cardiovascular risk factors of atherosclerosis, diabetes, and/or aging, trigger endothelial activation and subsequent endothelial dysfunction by promoting a pro-oxidant and pro-inflammatory environment. This in turn leads to the release of cytokines and endothelial microparticles, which further promote endothelial injury. Oxidative stress dysregulates endothelial nitric oxide synthase (eNOS) production, which leads to reduced nitric oxide (NO) bioavailability and impaired endothelial vasodilatory responses. Microthrombi may be formed on the endothelial surface. Endothelial disruption leads to loss of blood–brain barrier (BBB) integrity and tight junction damage, causing microvascular bleeding. The BBB is comprised of endothelial cells strengthened by tight junctions on the luminal side and a basement membrane with pericytes and astrocyte endfeet projecting onto the basement membrane. Endothelial dysfunction and BBB disruption causes astrocytes to detach from the BBB. Inflammation also causes the disruption of cell–cell interactions and degradation of the basement membrane, which exacerbates cell injury. This leads to a narrowing of the lumen and subsequent reduction in cerebral blood flow (CBF). This increases the probability of hypoxia and injury to neuronal cells and the parenchyma. **Ischemic stroke**: Occurs due to an occlusion in the cerebral blood vessels, which restricts blood flow in the brain. The occlusion is commonly caused by the formation of a thrombus after damage to the endothelium. Damage to cells causes the release of ATP and upregulation of vWF on the endothelium. ATP is converted to ADP, leading to platelet activation and aggregation to the site of injury, thus forming a thrombus. This triggers several downstream processes, which further dysregulates the endothelium. The lack of blood flow leads to immediate energy failure and excitotoxicity, which leads to an increase in intracellular Ca^2+^ and Na^+^. Increases in intracellular Ca^2+^ directly increase mitochondrial permeability and induce the expression of cytochrome C, phospholipase A2, and cyclooxygenase, which cause membrane degradation and the increase of free radicals and reactive oxygen species (ROS), leading to oxidative stress. ROS causes DNA damage and the increase of inflammatory mediators such as cytokines (TNFα, IL-1β), adhesion molecules (VCAM-1, ICAM-1), and chemokines (IL-8, MCP-1), which in turn leads to increased leukocyte infiltration and microglial activation, causing widespread inflammation and apoptosis. The increased intracellular Na^+^ leads to peri-infarct depolarization and the increase of extracellular glutamate and K^+^ as well as BBB damage and vasogenic oedema (created with BioRender.com).

**Figure 2 biomolecules-11-00994-f002:**
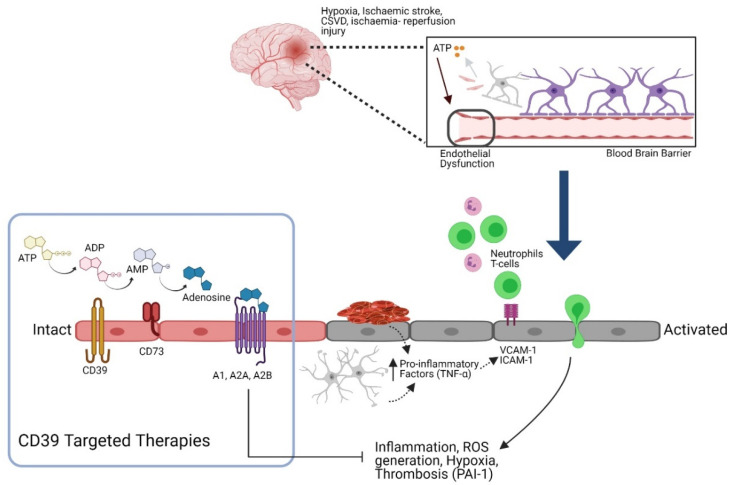
The role of the purinergic signaling pathway in regulating endothelial function. Hypoxia, thrombosis, and ischemia-reperfusion injury promote extracellular ATP (eATP) release from dying astrocytes and neurons that potentiates endothelial dysfunction. Damaged endothelial cells release pro-inflammatory factors such as TNF-α, which directly upregulate leukocyte-endothelial adhesion molecules (VCAM-1, ICAM-1) to facilitate leukocyte adherence and transmigration through the endothelium. This contributes to a pro-inflammatory and pro-oxidative environment. CD39 is expressed on endothelial cells in the vasculature and converts eATP and ADP into AMP. AMP is catabolized by CD73 to generate adenosine. While ATP and ADP potentiate inflammation and thrombosis, adenosine inhibits platelet aggregation, inhibits inflammation and oxidative stress (created with BioRender.com).

## Data Availability

Not applicable.

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
