# Peer review of "Role of Purinergic Signalling in Endothelial Dysfunction and Thrombo-Inflammation in Ischaemic Stroke and Cerebral Small Vessel Disease"

_biomolecules, 2021, doi:10.3390/biom11070994_

Round 1
Reviewer 1 Report
The authors reviewed the role of purinergic signaling in endothelial dysfunction and thrombo-inflammation in brain ischaemic diseases. Overall the article is well written. The topic is interesting and clearly discussed, the context is easy to follow.
Author Response
Thank you for your timely review.
Reviewer 2 Report
This review summarized functional and morphological changes of endothelium under the pathophysiological conditions such as ischemia and cerebral small vessel disease, and especially focused on the role of purinergic signalling in it. The content is simple and easy to understand. I have several minor concerns as follows.
Minor comments
1) P3, L108. The mechanisms underlying that decreased erythrocyte deformability decreases NO activity should be described with adequate references.
2) P3, L109 and P4, L163. The term “bioavailability” is inadequate for endogenous NO. “Bioavailability” is used for exogenous drugs.
3) P3, L133. What “change” do authors mean? Functional or morphological or others? The term “change” is too simple. Explain it more in detail.
4) P4, L167. Do authors mean “ The endothelium is further dysregulated” or “The endothelium further dysregulates something”?
5) P6 Figure 1. There are several cells other than endothelial cells (astrocyte, neuron ?) in the figure. These cells should be mentioned in the legend.
6) P6, L233. “subsequently and reduction” should be “subsequently reduction”.
7) P6, L241. Do the authors mean “induce the expression of …..”?
8) P6, L241. “Cytochrome C, Phospholipase A2, and Cyclooxygenase” should be “cytochrome C, phospholipase A2, and cyclooxygenase”
9) P7, L279. Are the references [69] and [70] correct? Looking at the contents, is the citation reversed?
10) P7, L290. What the term “In contrast” indicates?
11) P8, L306. CD39 does not convert ATP into adenosine. It should be “convert ATP/ADP into AMP”.
12) P10, L376. Do the authors mean “…….therapies include…..” or “………inhibitors are reported” or others ?
Author Response
Thank you for your review. Please see our itemised response in the attached document.
